# OpenReview forum: "Enhancing Adversarial Robustness of LLMs with Analytic Hierarchy Process"
_colmweb.org/COLM/2024/Conference — COLM_

### Official Review · Reviewer_cyeh · 2024-05-11

**Rating:** 6
**Confidence:** 3
**Ethics Flag:** 1

**Summary:**

- This paper suggests an adversarial defense method that automatically detects and purifies diverse types of adversarial noises based on LLM.
- Instead of detecting a single type of adversarial noise (e.g., word-level attack) by a detector model [1], the proposed framework iteratively classifies the type of adversarial noise and purifies that noise by LLM.
- Similar to the verbal reinforcement of Reflexion [2], this framework utilizes the history of iterations as the part of input context to improve the subsequent classification and refinement.
- The empirical results show that the proposed method could enhance the robustness of LLMs in various attack scenarios such as word-level attacks and jailbreak attacks.

[1] Detection based Defense against Adversarial Examples from the Steganalysis Point of View, Jiayang Liu et al., CVPR 2019.

[2] Reflexion: Language Agents with Verbal Reinforcement Learning, Noah Shinn et al., NeurIPS 2023.

**Questions To Authors:**

- **Clarification on Tables 1 and 2** The inferences with different defense methods lead to different outputs. Hence, for each defense framework, the attack method leads to different adversarial examples. In Tables 1 and 2, there can be two possibilities: 1) it evaluates the ASR of each defense system with adversarial examples to inference of standard model, or 2) it evaluates the ASR of each defense system with adversarial examples to inference of each defense system. Since the attacker can only access the API with the defense system, 2) seems more convincing. Can you explain the evaluation in more detail?
- **Effectiveness of each component of AHP** Let's assume the simple scenario with the word-level attack. The inspection step of the AHP alternates the role of the detector model and the refinement step of the AHP alternates the role of text purification methods. Under this view, one can alternate the refinement step of the AHP with purification methods such as Dirichlet neighborhood ensemble [3]. Also, one can alternate the inspection step of AHP with detection methods such as robust density estimation [4]. Moreover, the adversarial robustness can be evaluated with more strong attack methods such as [5]. Can you provide additional ablation studies that show the effectiveness of each component in AHP?
- **Performance of inspection step** It's just a curiosity question. Can we check the classification performance of the inspection step?

[3] Defense against Synonym Substitution-based Adversarial Attacks via Dirichlet Neighborhood Ensemble, Yi Zhou et al., ACL 2021.

[4] Detection of adversarial examples in text classification: Benchmark and baseline via robust density estimation, Kiyoon Yoo et al., ACL 2022.

[5] Query-Efficient and Scalable Black-Box Adversarial Attacks on Discrete Sequential Data via Bayesian Optimization, Deokjae Lee et al., ICML 2022.

**Reasons To Accept:**

Strengths
- The proposed method shows superior performance compared to the baseline methods.
- In contrast to the existing detector-based defense methods that require the training of detector models, the proposed method utilizes a pre-trained LLM without fine-tuning as the detector model (AHP engine) which can handle multiple types of adversarial noises.

**Reasons To Reject:**

Weaknesses
- The AHP engines are based on LLMs, which require expensive inference costs. Moreover, the AHP framework utilizes this expensive engine for several iterations (in paper, n=10).
- The proposed method adopts the verbal reinforcement framework appropriately to the problem of adversarial defense. However, we can also view the proposed AHP framework as an extension of the detection-based adversarial defense framework. It would be helpful if the paper shows the additional ablation studies with existing detection and purification methods.
- This paper handles the scenarios with various types of adversarial noises. However, each empirical result illustrates the robustness of models to the adversarial examples with a single type of noise.

---

> ### Author Rebuttal · Authors · 2024-05-31
>
> **R1:** Inference costs.
>
> **A1:** Thanks. Our method indeed increases inference costs compared with direct response to user input prompt. In general, one may either improve training-time alignment or inference-time alignment. Training-time alignment for LLMs is highly cost and hard to adapt to multiple attacks. Our method takes the inference-time alignment approach and is capable of adaptively defense against multiple attacks. As we do not increase training-time costs, and there is no free lunch for enhancing model robustness, it is inevitable to increase inference costs. However, our method can be used as a general and quick ad-hoc strategy to deal with new potential risks, and it can be promptly applied to any model.
>
> Besides, we have analyzed the trade-off between inference costs and the effectiveness of robustness of our framework in Appendix F (Ablation Study). In the ablation experiment, we gradually removed reasoning steps. As the removal of reasoning steps led to an improvement in reasoning efficiency, the model's robustness gradually decreased. The discussion of inference cost is in Appendix C. We set the upper limit to 10 iterations, and all models successfully completed the test before reaching the iteration limit.
>
>
> **R2:** Ablation
>
> **A2:** Thanks for the valuable suggestions. Our comparative methods can be viewed as simplified versions of traditional methods, such as spell-check and Paraphrase, both of which attempt purification. We are not necessarily striving to achieve state-of-the-art performance in every subtask. Instead, our primary objective is to provide a robust and adaptable solution for adversarial defense. The effectiveness of our proposed method has been demonstrated through comparative experiments.
>
> **R3:** Single type of noise
>
> **A3:** Thank you for suggesting a more challenging validation scenario. Our current choice of validating each type of attack individually is mainly due to the consideration of representative attack methods, all of which utilize a single type of noise. If there are new mixed attack methods, we will add them to the supplementary experiments. Additionally, we manually tested several handwritten mixed noises (Adversarial Noises mixed with Automatic Gradient). Our method remains effective in manual tests, which is primarily due to our defense framework not being targeted at a specific type of noise.
>
> (We have answered all the questions, but due to the maximum length limit can't provide them at this stage)

---

### Official Review · Reviewer_yKjv · 2024-05-11

**Rating:** 6
**Confidence:** 3
**Ethics Flag:** 1

**Summary:**

This paper proposes a framework to enhance the adversarial robustness of LLMs without training the LLM. Specifically, the proposed method decomposes the task into three steps: 1) Safety and Validity Assessment, which iteratively inspects and refines the input text; 2) Secure Response Synthesis, which infers and rectifies the output. Experimental results show that the proposed method obtains better results than several baselines on two benchmarks.

**Reasons To Accept:**

1. This paper is well-organized and easy to follow.
2. This paper introduces a universal method that can deal with complex adversarial attacks, rather than tailoring to specific attack scenarios. Moreover, the method introduced in this paper does not require training or optimization.
3. The proposed method gives better results than previous baselines.

**Reasons To Reject:**

1. The baselines compared in this paper are weak, there should be more recent baselines for comparison, such as [1][2].
2. The connection between the proposed framework and the Analytic Hierarchy Process (AHP) seems not clear, it would be better to clarify this.
3. The adversarial attack text used in this paper is somewhat weak, and it would be better to show how the model performs when testing on adversarial inputs generated by attacking LLMs such as [4].

[1] Defending ChatGPT against jailbreak attack via self-reminders, Nature Machine Intelligence, 2023
[2] LLM Self Defense: By Self Examination, LLMs Know They Are Being Tricked. ICML, 2023
[3] GradSafe: Detecting Unsafe Prompts for LLMs via Safety-Critical Gradient Analysis, arxiv, 2024
[4] An LLM can Fool Itself: A Prompt-Based Adversarial Attack, ICLR, 2024

---

> ### Author Rebuttal · Authors · 2024-05-31
>
> **R1:** The baselines.
>
> **A1:** The two defense methods you mentioned, [1] and [2], are specifically designed for jailbreak scenarios. Results in the table below.
>
> | Model | Method | Harmful Content  | Adult Content |  Illegal Activity  | Fraudulent Activity |
> |---|---|---|---|---|---|
> | LLaMA2-70B-Chat (70B) | Standard | 31.0 | 17.0 | 27.0 | 48.0 |
> |  | Self-Reminders | 3.0 | 0.0 | 4.0 | 7.0 |
> |  | Self-Defense | 3.0 | 0.0 | 3.0 | 8.0 |
> |  | AHP (Ours) | 2.0 | 0.0 | 2.0 | 2.0 |
> | GPT-3.5-Turbo (176B) | Standard | 17.0 | 29.0 | 37.0 | 41.0 |
> |  | Self-Reminders | 3.0 | 7.0 | 11.0 | 6.0 |
> |  | Self-Defense | 2.0 | 6.0 | 12.0 | 7.0 |
> |  | AHP (Ours) | 1.0 | 0.0 | 0.0 | 0.0 |
>
> Our method can handle a variety of different attacks and provides a more detailed verification of the defense, enhancing the robustness of chat models by iteratively checking for potential risks and verifying output safety. When modules in our method are simplified or removed, AHP can degrade to either [1] or [2]. Therefore, our results consistently outperform [1] and [2].
>
> **R2:** Connection.
>
> **A2:**
> We have described the connection between the proposed framework and AHP in Introduction Section. The AHP, developed by Thomas L. Saaty, is a decision-making process that uses a multi-level hierarchical structure of objectives, criteria, sub-criteria, and alternatives. The main principle of AHP is that it decomposes a complex problem into a hierarchy of more easily comprehensible sub-problems, each of which can be analyzed independently. In the context of our work, we have adapted the AHP framework to guide the decision-making process of LLMs. The complex task of defending against diverse and potentially malicious inputs is broken down into smaller, more manageable sub-tasks. These sub-tasks are then prioritized and systematically addressed, similar to how the AHP breaks down decision-making into a hierarchy of criteria and alternatives.
>
> **R3:** Attack[4].
>
> **A3:**  We have supplemented the experiment on [4]. (Due to length limit, see table in response to Reviewer Teap). It can be seen that our method still exhibits strong robustness against this new type of attack. This is because our method is not designed for specific tasks, but rather, it gradually eliminates noise through multiple rounds of understanding language via LLMs, therefore it has good generality.
>
> We highly appreciate your valuable feedback. We hope these answers will help you re-evaluate the rating score of the paper.

---

> > ### Comment · Reviewer_yKjv · 2024-06-04
> > **Thanks for your response**
> >
> > The experimental results have addressed my concerns, and I raise my score.

---

> > ### Author Response · Authors · 2024-06-05
> > **Thanks for your valuable review**
> >
> > We sincerely thank the reviewer for the valuable review and acknowledgement of our work.
> >
> > Best,
> >
> > Authors of paper 343

---

### Official Review · Reviewer_UATu · 2024-05-13

**Rating:** 6
**Confidence:** 3
**Ethics Flag:** 1

**Summary:**

This paper proposes a new prompting process that enhances adversarial robustness of LLMs. The problem of defending adversarial attacks of LLMs is an important and emerging area. The proposed process leverages the power of the target LLM by asking it to inspect, refine, and compare the input text in multiple rounds in order to generate a refined text with less adversarial noises. As a prompting method, the proposed method can be used in enhancing the robustness of any LLM. The paper empirically shows the effectiveness of the proposed method with comprehensive experiments. As the method needs to do multiple rounds of queries, it adds extra complexity to the inference.

**Questions To Authors:**

How are the instructions of the steps (Inspect, Refine, Compare, Evaluator, Verification) determined and how to make sure that they are the optimal prompts? Similarly, how are the instructions of task inference for different models determined?

**Reasons To Accept:**

1. The proposed method is motivated, intuitive and shown effective empirically.

2. The paper is clearly written and easy to follow.

3. The experiments are comprehensive.

**Reasons To Reject:**

1. The proposed one is a heuristic method without theoretical analysis. It is unclear that how well the method can generate other than the LLMs and datasets used in the paper.

2. The performance of the proposed method shown in the main results of Table 1 seems to be a bit fluctuating. Except for the model of GPT-3.5-Turbo, there is no clear win of the proposed method for the other models. Although the proposed method achieves better performance on average.

---

> ### Author Rebuttal · Authors · 2024-05-31
>
> **R1.** Theoretical analysis
>
> **A1:** We have designed a general framework to cope with various potential adversarial attack risks. Similar to most LLM Agent works, we cannot provide theoretical guarantees. Our contribution is to divide the safety reasoning process into two objectives. The part that ensures safety is designed as multi-round iteration. When one round cannot complete the correction of noise, multi-round iteration can increase the possibility of identifying and correcting the noise. Table 6 shows the number of rounds needed for different models.
> For scalability on more models and datasets, we used 4 representative large models including llama2 and Chatgpt-3.5 in the experiment, and verified under 3 different attack methods. The experimental results show that there is an improvement in robustness on all models. On the other hand, our reasoning framework is a general framework, not specifically designed for specific noise, so it has good scalability.
>
> **R2.** Table 1 results
>
> **A2:** Thank you for your question. The experimental results in Table 1 are mainly due to AHP uses the LLM itself to complete the safety check, which requires the model to have strong semantic understanding capabilities. Therefore, when the language model capabilities are limited, the effect is not obvious. Besides, we verified the effect of using different models as the AHP Engine (Figure 3). When a stronger model is used, our method shows more obvious improvement.
>
> **Q1.** Instruction design
>
> **Answer:** Thanks. Generally speaking, it's hard to claim which prompt is optimal. Even for system prompts of LLMs, they are usually hand-crafted. On the other hand, as the instruction-following ability of LLMs increases, the need to search for an optimal prompt becomes less. For the instructions in our method, we conducted iterative manual tuning according to the objectives of each subtask, and finally found a good instruction. The experiment proved that the designed instruction is effective. We cannot guarantee that this is the optimal instruction, but if the optimal one can be found, our method can achieve better results. As for the instructions of the downstream tasks, we have adopted the settings of previous works [1,2] for the sake of experimental consistency.
>
> [1] Wang, Jindong, et al. "On the robustness of chatgpt: An adversarial and out-of-distribution perspective.".2023
>
> [2] Liu, Yi, et al. "Jailbreaking chatgpt via prompt engineering: An empirical study.".2023

---

> > ### Comment · Reviewer_UATu · 2024-06-06
> >
> > Thank the authors for the replies. I'm happy to keep my current positive score of the paper.

---

> > > ### Author Response · Authors · 2024-06-06
> > > **Thanks for your valuable comment**
> > >
> > > Thanks for your comment. We appreciate your time and effort in reviewing our work.
> > >
> > > Best,
> > >
> > > Authors of paper 343

---

### Official Review · Reviewer_Teap · 2024-05-14

**Rating:** 7
**Confidence:** 2
**Ethics Flag:** 1

**Summary:**

The paper proses a framework for LLM inference that aims at improving the adversarial robustness of the output. Specifically, the framework Analytic Hierarchy Process or AHP consists of multiple subtasks (1) inspect to provides textual feedback for the refinement of the input text (2) refinement is a iterative process with the LLM to remove noise tokens. This is followed by a secure response synthesis step to produce the response at the end. The proposed framework shows robust performance versus a few adversarial attacks across several benchmarking datasets. This is certainly an important topic in the field of Safety and Trustworthiness of LLMs.

**Ethics Concerns Details:**

No concerns

**Questions To Authors:**

- Have the authors verified or done the sanity check on the data contamination issue when evaluating the framework?
- Do you have any human evaluation studies to verify the findings?

**Reasons To Accept:**

Overall, the proposed framework is intuitive, justifiable, and easy to understand. The results on benchmarking datasets are also promising.

**Reasons To Reject:**

The evaluations are based on benchmarking datasets that are released on the web. I am wondering if study considered the test data contamination issues where the LLM models have seen the datasets in their training phases. Also, it is also better to include human evaluation besides benchmarking datasets to state the robustness of the proposed framework.

---

> ### Author Rebuttal · Authors · 2024-05-31
>
> We thank the reviewer very much for the acknowledgment of our work and valuable feedback on further improving this work.
>
> **R1:** The evaluations are based on benchmarking datasets that are released on the web. I am wondering if study considered the test data contamination issues where the LLM models have seen the datasets in their training phases.
>
> **A1:**
> Thank you for raising this insightful concern. We have checked the knowledge cut-off dates of the models used in our experiments to avoid data contamination. The cut-off date for Llama2 is September 2022, and for Chatgpt-3.5, it's September 2021. The adversarial samples we used, such as AdvGLUE and GPTFUZZER, were released in November 2021 and October 2023, respectively. The Automatic Gradient Generated samples were dynamically generated. Therefore, we believe that there were no data contamination issues in our experiments. Additionally, we have supplemented our study with the latest attack[1] methods as further verification. (due to max length limit, only gpt-3.5 results are show below)
>
> | 　 | Task | SST-2  | QQP |  MNLI-m |  MNLI-mm |  RTE | QNLI |  Avg |
> |---|---|---|---|---|---|---|---|---|
> | GPT-3.5 | PromptAttack-EN | 56.00 | 37.03 | 44.00 | 43.51 | 34.40 | 40.39 | 42.54 |
> |  | AHP (ours) | 51.61 | 30.81 | 39.83 | 36.92 | 27.82 | 33.40 | 36.22 |
> |  | PromptAttack-FS-EN | 75.23 | 39.61 | 45.97 | 44.10 | 36.12 | 49.00 | 48.34 |
> | 　 | AHP (ours) | 70.74 | 34.44 | 40.24 | 39.71 | 31.56 | 43.91 | 43.35 |
>
> The results(ASR) above are consistent with the conclusions of our existing experiments. The effectiveness of our method stems from the large language models' understanding of language semantics. Through our designed APH framework, LLMs can autonomously uncover potential risks.
>
> **R2:** Also, it is also better to include human evaluation besides benchmarking datasets to state the robustness of the proposed framework. Do you have any human evaluation studies to verify the findings?
>
> **A2:** Thanks. Human evaluation indeed enhances the comprehensiveness of the experiment. In our experiments, we already have attack methods built with manual templates, which can be seen as human verification. The results are shown in Table 2. The experimental results on manual template attacks have proven the effectiveness of the method. For human evaluation of defense process, we have provide case studies in Appendix F.
>
> [1] An LLM can Fool Itself: A Prompt-Based Adversarial Attack, ICLR, 2024

---

> > ### Comment · Reviewer_Teap · 2024-06-04
> >
> > I have read the answers and updated my score accordingly.
> > Overall, the proposed framework is interesting and shows promising results. Although, I would still recommend further analysis to make sure that the performance is reproducible. The authors can explore the working of AHP on other dimensions of trustworthiness.

---

> > > ### Author Response · Authors · 2024-06-05
> > > **Thanks for your valuable comments**
> > >
> > > Many thanks to the reviewer for the valuable comments and the acknowledgement of our work. We shall seriously address the reviewers' comments on evaluation and reproducibility, and provide human evaluation results as well as other further analysis in the revised paper.
> > >
> > > Best,
> > >
> > > Authors of paper 343

---

### Decision · Program_Chairs · 2024-07-10

**Decision:**

Accept

**Comment:**

This paper proposes a framework to improve adversarial robustness at inference time. The process involves iterative processes of inspecting and refining generated text.

Overall reviewers are all positive about this paper. They appreciate that the method is intuitive and well-motivated, and that the empirical results are convincing of the effectiveness of the method. They also appreciate that the paper is clear and well-written, and that the method is general (not attack-specific) and doesn't require fine-tuning.

Reviewers raise some concerns, including that the method adds computational cost, and that there could be more theoretical analysis, incorporation of additional stronger baselines and attacks, and possible additional ablations. however, on the whole authors have satisfactorily addressed these concerns in their responses, and this seems like a solid submission for publication.

[comments by the PCs] Please follow up on the AC recommendations. Discussing limitations and costs can improve the paper and provide a more complete picture of the approach.